# Sports Supplements User Profile Based on Demographic, Sports, and Psychological Variables: A Cross-Sectional Study

**DOI:** 10.3390/nu14214481

**Published:** 2022-10-26

**Authors:** Leticia Mera-Zouain, José Luis Carballo, Mercedes Guilabert Mora

**Affiliations:** 1School of Psychology, Pontificia Universidad Católica Madre y Maestra (PUCMM), Autopista Duarte, Km 1 1/2, Santiago de los Caballeros 51000, Dominican Republic; 2Center of Applied Psychology, Miguel Hernández University, Comunidad Valenciana, 03202 Elche, Spain; 3Psychology Health Department, Miguel Hernández University, Comunidad Valenciana, 03202 Elche, Spain

**Keywords:** sports supplements, prevalence, sports, sports motivation, sports dependance, muscle dysmorphia

## Abstract

Despite the high prevalence of sports supplement (SS) use, efforts to profile users have not been conclusive. Studies report that 30–95% of recreational exercisers and elite athletes use SS. Research found has mostly focused on demographic and sports variables to profile SS users, but little research has studied the psychological factors that may influence the use of SS. The purpose of this investigation was to classify, describe, and differentiate the profile of users and non-users of SS, considering demographic, sports, and psychological variables. A total of 554 participants completed the questionnaire. Overall, 45% of recreational exercisers and elite athletes reported using supplements. There were significant differences found regarding the use of SS between men and women (51% vs. 49%, *p* = 0.002; OR = 1.799), and when training 4 or more days per week (*p* ≤ 0.001; OR = 1.526). Findings regarding the psychological variables have been found in the Adonis Complex. These results indicate that participants with greater concerns regarding physical appearance, tend to be SS users (*p* = 0.001; OR = 1.200). The results of this study fill a gap in previous research, and provide an approximate profile, including demographic, sports, and psychological variables of SS users.

## 1. Introduction

The use of sports supplements (SS) is widespread among professional exercisers, also known as elite athletes, that is, athletes at different competitive levels [1,2,3], and in recreational exercisers, that is, those who attend gyms or exercise regularly [1,2,3,4,5,6,7]. SS are commercially available products that have been defined and categorized in many ways [3,4,5]. In general, SS include pills, powders, drinks, or gels used for bodybuilding, increasing energy, weight loss, and enhancing performance [4,5,6]. The International Society of Sports Nutrition and The International Olympic Committee of Sports Supplements have grouped SS into five different categories: ergogenic supplements, medical supplements, functional foods, sports foods, and other supplements, referring to herbal and botanical extracts [5,8,9].

The prevalence of SS use among elite athletes and recreational exercisers has been studied by different authors and has greatly expanded throughout the years [2,4,8,9,10,11,12,13,14,15,16,17,18,19,20,21,22,23,24,25,26]. The range of SS users that has been reported in previous investigations is very wide [1,2,3,4,5,7,8,10,11,12,13,14,15,16,17,18] and goes from 35 to 95% [5,6,7,8,17,18]. According to recent literature, this high prevalence of SS use may become dangerous for health [17,18,19] due to the misuse, lack of reliable resources of information, and the scarce guidance of health specialists [21,22,23,24,25,26,27,28].

Previous studies have investigated the demographic profile of SS users, regarding sex and age [11,17,18,22,23,24,25,26,27,28]. These researchers have reported sex-based and age differences in the distinctive subpopulations studied [23,24,25,26,27,28,29,30,31,32,33]. Sex-based differences have been considered an important area of debate [25,32,33,34], where some authors have reported that men tend to use more SS [7,8,22,25,27], and others have suggested a greater prevalence of SS use in women [8,29,30]. Concerning age, authors have informed that SS use starts at a young age and continues into adult life [8,9,10,12,18,21]. Furthermore, they have reported that due to competition and nutritional needs, elite athletes tend to use more SS than recreational exercisers throughout their life [3,8,20,21,22,23,24,25,33,34,35].

However, recent research has informed the importance of studying other variables that may profile SS users [6,7,8,9,10,30,31,32,33]; preliminary findings indicate that the users of SS should be studied underlying the motivation of use and psychopathological illnesses that may be present [36,37]. According to Mudrack et al. [36], who studied Sports Motivation (SM) (i.e., Self-determination in sports), there is a possible correlation between SM and the use of SS or enhancing products (i.e., SS that may contain doping agents, such as oxilofrine, which may be found in fat burners), where participants with SM and anxiety were more likely to use SS [34,36]. In addition, the literature has reported that severe obsessive-compulsive behaviors, such as Muscle Dysmorphia (MD) (i.e., variant of body dysmorphic disorder) and exercise abuse, may also influence in the decision of using SS [35,36,38,39,40,41,42].

Recent literature has studied the correlation between exercise abuse, MD, appearance anxiety, and compulsive behavior [36,40]. Findings have suggested a high risk of exercise abuse and body image disorders in recreational exercisers that used fitness products under no supervision [36,37,40,41,42,43,44,45,46,47,48,49,50,51]. Moreover, they have informed that changes in habits, such as radical diets and the use of SS, have been reported as risk factors of MD, exercise abuse, and addictive behaviors in exercisers [34,35,36,38,39,43,44,45,46,47,48,49]. Therefore, authors have reported users of SS with high levels of SM, MD and exercise abuse which are vulnerable to a risky intake that may lead to physical and mental problems [36,37,42,43,44,45,46,47,48,49]. These authors have indicated that these findings are not conclusive and suggest new research is needed to be able to establish a profile of SS users [35,36,37,38,39,43,44,45,46,47,48,49,50,51].

This study aims to not only to describe the demographic and sports-related characteristics of SS users, but also to incorporate SM, exercise abuse, and MD analyses, to integrate a psychological perspective in the evaluation of the profile of SS users. Therefore, this study aimed to identify the sociodemographic characteristics and anthropometric measures of users of SS (USS) vs. non-users of SS (NUSS). Second, we aimed to establish the differences between USS and NUSS in terms of sports-related variables, such as status (being an elite athlete vs. a recreational exerciser), days of training per week, hours of exercise per day, the preferred time of the day for training, and the psychological variables (SM, exercise abuse and MD). Finally, we aimed to be able to profile the USS within demographic, sports and psychological variables, a classification model was developed.

## 2. Materials and Methods

### 2.1. Study Design

This study was approved by the Committee of Research and Ethics of the Miguel Hernández University of Elche (reference number: DPS.JCC.03.19) and was conducted in 2020–2022. In this descriptive cross-sectional study, exercisers were recruited in December of 2020/2021.

The methodology of this article was in accordance with the STROBE statement [52].

### 2.2. Participants

To calculate the minimum sample size, Sampsize program [53] was used. Since the exact prevalence of SS has not been established, and the estimations range from 30% to 95%, for this study we calculated the size of the sample with a prevalence of 50%, with a 5% margin of error and a 95% confidence level. The minimum sample required was 385 participants. The inclusion criteria for our study were as follows: (1) signing informed consent; (2) participants had to be recreational exercisers, i.e., participants that were physically active but not at a competitive level, or elite athletes, i.e., participants that were physically active at a competitive level; and (3) Spanish speaking. The only exclusion criteria were not completing the questionnaire correctly. All the participants signed a consent form before participating in the study.

During the study, 565 participants had access to the questionnaire, but only 554 participants met the inclusion criteria. See Figure 1 for the details of the study participants. Participants were recruited through a two-phase sampling design via WhatsApp, Email, Instagram, and Facebook. To avoid over- or under-representation of any specific group in the sample and to minimize the bias of non-probability sampling, 20 initial participants (“seeds”) were selected to initiate the survey link distribution by a snowball sampling method [54,55]. Seeds were selected purposely to be diverse in age and were contacted via WhatsApp. All participants were directed to a survey link, which was created at the beginning of the research. This survey link directed them to the informed consent, which had to be accepted to be able to continue. The average time to complete the questionnaire was 15 min. In the second stage, the survey link was sent out widely via WhatsApp and Email and was posted on Instagram and Facebook.

Table 1 shows the main demographic, sports characteristics, and use of SS. The sample population (*N* = 554) comprised of 317 women (57%). The mean age was 28.02 ± 11.18 (ranging from 18 to 68 years). The range mean for BMI was 24.06 ± 4.52. Regarding the sports characteristics, most participants were recreational exercisers, 72% (*n* = 399). The mean of their training days was 3.97 ± 1.289, with a time of training per day mean of 1.66 ± 0.83. Most frequently, the participants preferred training in the morning, 62% (*n* = 342). Regarding the use of SS, 45% were USS (*n* = 249).

### 2.3. Measures

The primary outcome variable was the use of SS, which was assessed with a self-administered online form. This form was composed of seven different sections: (1) informed consent, (2) demographic and anthropometric details, (3) physical activity pattern, (4) SS use, (5) Sports Motivation Scale [51], (6) Sports Addiction Scale [56], and (7) Adonis Complex Questionnaire [57].

Demographic information included age and sex. Anthropometric measures included weight in kilograms and height in centimeters, which were both needed to calculate the BMI (Body Mass Index). Furthermore, the physical activity questionnaire assessed sports status (recreational exerciser or elite athlete), days of training per week, hours of training per day, preferred time of the day for training, and the type of sport or exercise practiced. Afterwards, participants were asked: “Do you use SS?”. Participants that self-reported using SS were asked to identify the frequency of use and type of supplement.

SM was measured with the Spanish version of the Sports Motivation Scale (SMS) from Balaguer and Duda [51]. This scale consists of 28 questions with a 7-point Likert scale: it has nothing to do with me (1, 2), it has something to do with me (3–5), or it completely has to do with me (6, 7). The purpose if this scale is to measure three types of Intrinsic Motivation (IM), IM to know, IM to experiment, and IM to achieve results; three types of Extrinsic Motivation (EM), EM for external regulation, EM for introjected regulation, and EM for identified regulation; and Amotivation [51]. Previous studies have confirmed its internal consistency (*α* between 0.61 and 0.88) [51]. Our study also showed an acceptable internal consistency (*α* = 0.91).

We measured sports addiction via the Sports Addiction Scale 15 (SAS-15) [56], which is the reduced 15-item version of the SAS-40. The following five factors were measured: F1 “dependence”, F2 “lack of control”, F3 “loss of interest”, F4 “continuity”, and F5 “concern”. Previous studies have confirmed an adequate internal consistency (*α* = 0.84) [56], which was also the case for our study (*α* = 0.83).

The risk of Adonis Complex, MD, or bigorexia was measured with the “Adonis Complex Questionnaire” (ACQ) [57]. The ACQ consists of 13 items, in which the subjects are asked to choose between 3 possible answers. These answers indicate in increasing order the presence and severity, from normal to pathological, of the concern about physical appearance and how it affects the responder’s personal and social life. Values from 0 to 9 indicate minor concern, 10 to 19 indicate mild to moderate, 20 to 29 serious concern, and 30 to 39 severe forms of body image dissatisfaction and MD [57]. The internal consistency of this questionnaire in other studies has been acceptable (*α* = 0.88) [57], as well as in ours (*α* = 0.75).

### 2.4. Statistical Analysis

Statistical analyses were conducted using Statistical Package for the Social Sciences (IBM) v26 for MAC. Statistical significance was determined at *p* < 0.05. Descriptive and frequency statistics were used to describe the sociodemographic and anthropometric profile, physical activity status, and SS use questions. Data normality was tested by using the Shapiro–Wilk and Kolmogorov–Smirnov tests. The chi-squared test was used as the contrast statistic for categorical variables, and Student’s *t*-test was used for continuous variables. Phi was used to calculate effect size for categorical variables and Cohen’s d were used for effect sizes of the continuous variables. For Cohen’s d, 0.20 was small, 0.50 was medium, and 0.80 was large [55]. For Phi, 0.20 was weak, 0.40 was moderate, 0.60 was relatively strong, 0.80 was strong, and 1.00 was very strong [55].

Due to the usefulness of the Binary Logistic Regression for modeling the dependance of a binary response variable on one or more explanatory variables and the sample size for this study [55,58], this analysis was able to determine variables that classified the USS. The SS variable was dichotomized as USS and NUSS, the variables included in the model were demographic (age and sex), sports (days of training, hours of training per day, and preferred time of the day for training), and psychological variables (SM, exercise abuse, and MD) that showed statistically significant differences in the bivariate analysis.

## 3. Results

### 3.1. Differences between USS and NUSS in Demographic and Sports Variables

Overall, 45% (*n* = 249) of the participants were SS users. In this study, significant differences within USS and sex were found, where men tended to use more supplements than women (χ^2^ = 8.092; *p* = 0.004) with a small effect size (Φ = 0.121). The mean age of USS was 28.45 ± 10.69. BMI (24.21 ± 4.71) of the participants in this study showed no relationship with the use of SS. No differences were found regarding recreational exercisers and elite athletes within the use of SS. Participants that trained more days were more likely to use SS (*t* = 7.024; *p* < 0.01; d = 0.599). USS trained at least four days a week for a mean time per day of 1.73 ± 0.76 *h* and preferred training in the morning (65%, *n* = 161) (Table 2).

### 3.2. Differences between USS and NUSS in SM, Exercise Abuse, and MD

#### 3.2.1. Differences between USS and NUSS in Sports Motivation

In this study, significant differences within USS and sports motivation were found, where USS scored higher than NUSS in the following factors: EM for introjected regulation *t* = 3.582; *p* < 0.01) with a medium effect size (d = 0.308), IM to know (*t* = 2.566; *p* = 0.011) with a medium effect size (d = 0.220), IM to experiment (*t* = 2.726; *p* = 0.007) with a medium effect size (d = 0.233), and IM to achieve results (*t* = 2.624; *p* = 0.009) with a medium effect size (d = 0.260) (Table 3).

#### 3.2.2. Differences between USS and NUSS in Exercise Abuse

In this study, significant differences within USS and exercise abuse were found, where USS scored higher than NUSS in the factors of “dependence” (*t* = 3.456; *p* < 0.01) with a medium effect size (d = 0.252) and “concern” (*t* = 3.099, *p* = 0.002) with a medium effect size (d = 0.259) (Table 4).

#### 3.2.3. Differences between USS and NUSS in Adonis Complex

In this study, significant differences within USS and Adonis Complex were found, where USS scored higher than NUSS in the factor control of physical appearance (*t* = 5.287; *p* < 0.01) with a medium effect size (d = 0.45) (Table 5).

### 3.3. Binary Logistic Regression

Binary logistic regression yielded a significant model to determine the variables that classified USS. This model correctly classified 66.8% of the USS with a significant Chi-square result (*X*^2^ = 78.595, *p* < 0.01). The r^2^ of Nagelkerke was low (r^2^ = 0.177). As shown in Table 6, the results indicated that the set of variables that have contributed to classify positively the profile of USS are: male sex (OR = 1.799, CI 95%:1.242–2.604; *p* = 0.004), more days per week of training (OR = 1.526, CI 95%:1.310–1.778; *p* < 0.001), and higher levels of Adonis Complex regarding physical appearance concerns (OR = 1.200, CI 95%:1.076–1.338; *p* = 0.001).

## 4. Discussion

The present study aimed to investigate the prevalence of SS use, describe the sociodemographic and sports profile of SS users, and examine possible relationships between the use of SS and SM, exercise abuse, the Adonis complex, and MD. The current investigation was conducted according to the suggestions of Knapik et al. [6,7], using similar questionnaires (i.e., demographic and sports variables) and including the definition or examples of SS in the questions corresponding to supplement use. Psychological variables were included regarding the suggestions of Mudrack et al. [36] and Corazza et al. [40] regarding the motivation and psychopathological illnesses that may be related to USS.

According to our results, 45% of recreational exercisers and elite athletes reported to use SS, similar to the rate reported by other studies [1,2,3,6,7,8,9,10,11,12,13,14,15,16,17,18]. This fact should be considered with caution due to the underreporting of SS [2,3,4,5,7,8,11,12,13,14,15,16,17]. This underreporting is usually related to inaccuracies in users’ perception as well as the difficulty in admitting the use [16,17]. Regarding the demographic profile of SS users, results in our study showed that men tended to use more SS than women. This variable showed an important power of classification in the regression model (OR = 1.799). This finding that the participating men used more supplements than the women mirrors the results of previous studies [7,8,16,17,18,20,27,33,34]. The median age of USS in this study was 28 years (28.48 ± 10.65), similar to the age reported by other authors [2,8,23,24,25,26,27,28,29,30,31,32,33,34,35].

Compared to other studies [1,2,3,4,5,6,7,8,9,10,11,12,13,14,15,16,17], we did not find being an elite athlete or a recreational exerciser to be a significant predictor of SS use. When comparisons were made in by different researchers’, elite athletes were most likely to use SS in a greater extent recreational exercisers [7,8,9,10,11,17,18]. However, this result has been considered inconclusive by other authors, due to the different subpopulations studied and the lack of defining athletes or exercisers [7,9,11,12,13,14,15,16,17,19,20,21,22,23,24,25]. USS in our sample were more likely to train 4 or more days a week. This variable showed an important power of classification (OR = 1.526), which is consistent with what other authors have reported regarding the days of training per week [7,9,11,12,13,14,15,16].

The main novelty of this research resides in studying the relation of SM, exercise abuse and MD within USS. Regarding SM, IM and EM were analyzed. Bivariate analysis findings for EM have shown significant differences were found when USS scored higher than NUSS only in the factor of introjected regulation (*t* = 3.582; *p* < 0.01). For external and identified regulation, no significant differences were found. For IM, significant differences were found in the three factors analyzed: IM to know (*t* = 2.566; *p* = 0.011), IM to experiment (*t* = 2.726; *p* = 0.007), and IM to achieve results (*t* = 2.624; *p* = 0.009). These results contradict what previous research has reported, finding a negative relationship in general for the factors of IM towards the use of SS and a positive relationship of in the factors studied for EM towards a more positive attitude of being USS, also showing a strong correlation (*r* = 0.513) [32]. This previous study [36] has informed that athletes high in external motivation are more likely to use SS. We argue that the differences found in these studies may be a consequence of not specifying the characteristics of their participants, referring to them only as athletes that have goals and not stating whether they are elite or recreational [35,36,39]. Previous research has argued that higher scores in EM may be a result of the competitiveness of participants [36], which may be consistent with the results found for our study due to our sample, where most of the participants are recreational exercisers. Regarding the regression model, where SM did not contribute to classify the profile of USS, no previous research was found for running a logistic model that classifies the profile of USS.

Furthermore, when analyzing exercise abuse within USS vs NUSS, significant differences were found with higher scores in the factors of dependence (*t* = 3.456; *p* < 0.01) and concern (*t* = 3.099, *p* = 0.002). These results agree with those of other studies, where overall participants scored higher when analyzing exercise abuse and the use of supplements (*f* = 75.89; *p* < 0.001) [35]. When running the classification model, for our study, no significant differences were found regarding this variable, which differs from what Corazza et al. [42] found. In their research, exercise abuse was considered a strong predictor in the groups of USS (*p* < 0.001; OR = 3.03), even though they argue these results may be influenced by the competitiveness of the athletes’ subject of their study [42,43]. This may explain why in our study, exercise abuse was not considered a variable that classifies the profile of USS.

In addition, when analyzing the ACQ in USS vs NUSS, overall participants were concerned about their physical appearance; significant differences were found in the bivariate (*t* = 5.287; *p* < 0.01). These results are consistent with other studies. When studying MD, significant differences were also found regarding USS vs NUSS (*f* = 17.99; *p* < 0.001). For our study, when running the regression model, this variable was considered to classify the profile of USS (OR = 1.200). Indeed, as Khorramabady [59] has informed, participants who are subject to use SS, have reported to use them as an attempt to become less concerned about their physical appearance, due to the effects these products have in their body, (i.e., promoting muscle growth), and for acting as a fat burner or as an appetite suppressants [5,8,10,12,14,16,17,50,51,56,57,59,60].

The following limitations should be considered in this study. This research employed a cross-sectional design, which limited the classification of USS and did not allow us to draw conclusions about the causal relationships between the variables [55]. In this sense, further studies can consider employing a longitudinal study. Representativeness of the sample in terms of cultural background and diversity could also be improved by using a random selection method [58]. Moreover, this study was carried out during the COVID 19 pandemic and results may have been biased due to the lockdowns. Furthermore, even though we used a strategy of “seeds” to limit bias, there are limitations inherent to the method used, such as using self-report questionnaires that may cause fear of reporting to be USS and the estimated prevalence of SS use may be misreported either intentionally or due to misunderstandings; the lack of introspective ability, where the subjects may not be able to assess themselves accurately; and the interpretation of questions, where the wording may be confusing for participants and a social desirability bias, where participants might be prone to give answers which they consider are most sociably acceptable [61].

## 5. Conclusions

Finally, the present study suggests a model that profiles SS users reporting that being men, training four days or more a week, and having more concerns regarding physical appearance classifies positively the USS. Results found in this research seem to add knowledge regarding the profile of SS users by reporting a SS user profile regarding demographic, sports, and psychological variables all together. This research may be considered as a starting point to guide the development of health programs for exercisers and athletes interested in SS use. Furthermore, the findings will also be useful when creating educational and health awareness programs.

In addition, new research should consider the timeframes that describe the use of SS, the type of supplement used, motivations for SS use, and the effects these supplements may have in the participants. Moreover, future studies should confirm the analyses made in this investigation to be able to develop strategies to prevent the misuse of SS. To be able to confirm this predictive information investigators should also consider a structured interview with health professionals that will allow to classify and identify participants who are in more need of psychoeducation and intervention in terms of SS use more effectively. In general, the findings of our study may help to motivate the authorities to create regulations for the use of SS and this new knowledge provides information that contributes to create educational programs for the targeted population, being recreational exercisers or elite athletes.

## Figures and Tables

**Figure 1 nutrients-14-04481-f001:**
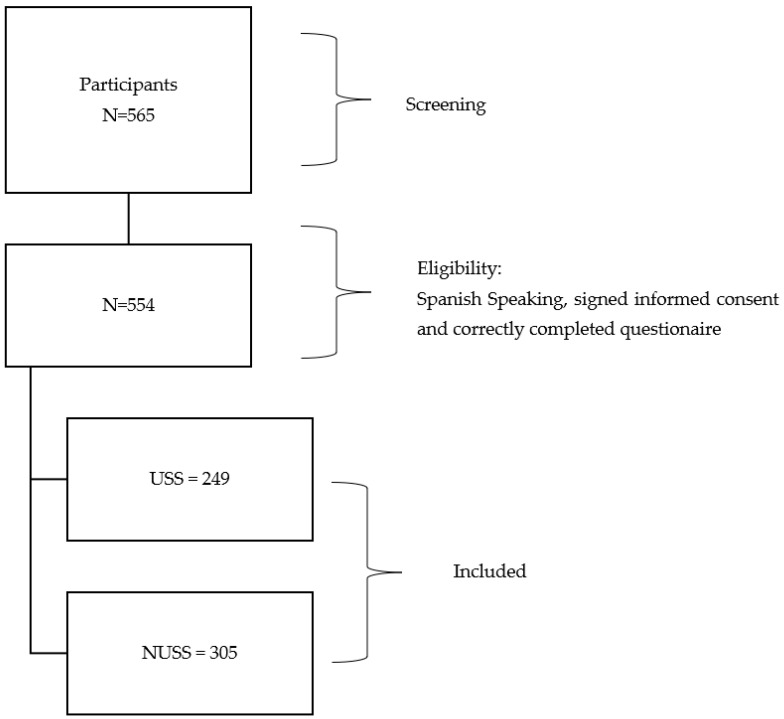
Flow chart detailing the study participants. USS, Sports Supplement Users; NUSS, Non-users of Sports Supplements.

**Table 1 nutrients-14-04481-t001:** Descriptive characteristics of the sample population (*N* = 554).

Variables	
Sex	
Women	57% (*n* = 317)
Men	43% (*n* = 238)
Participant age	
Mean ± SD (years)	28.02 ± 11.18
Range (years)	18–68
Sports status	
Professional	28% (*n* = 155)
Recreational exercisers	72% (*n* = 399)
Days per week	
Mean ± SD (days)	3.97 ± 1.289
Hours per day	
Mean ± SD (hours)	1.66 ± 0.83
Period of time	
A.M	62% (*n* = 342)
P.M	38% (*n* = 212)
Supplement use	
USS	45% (*n* = 249)
NUSS	55% (*n* = 305)

AM, Ante meridiem; PM, Post Meridiem; USS, users of SS; NUSS, non-users of SS.

**Table 2 nutrients-14-04481-t002:** Differences between USS and NUSS in demographic and sports variables.

Variables	USS (*n* = 249)	NUSS (*n* = 305)	*X^2^/t* (*p*-Value)	ES
Age, M ± SD	28.48 ± 10.65	27.63 ± 11.61	0.885 (0.377)	0.076 ^a^
Sex *n* (%)				
Women	126 (40)	191 (60)	8.092 (0.004) *	0.121 ^b^
Men	123 (52)	114 (48)		
BMI, M ± SD	24.21 ± 4.71	23.94 ± 4.37	0.693 (0.489)	0.060 ^a^
Days per week, M ± SD	4.38 ± 1.23	3.64 ± 1.24	7.024 (<0.001) *	0.599 ^a^
Hours per day, M ± SD	1.73 ± 0.76	1.60 ± 0.88	1.799 (0.073)	0.157 ^a^
Period of time, *n* (%)				
AM	161 (65)	181 (60)	1.639 (0.054)	0.054 ^b^
PM	88 (35)	124 (40)		
Sports status, *n* (%)				
Recreational exercisers	174 (70)	225 (74)	1.030 (0.310)	0.043 ^b^
Elite athletes	75 (30)	80 (26)		

M: mean, SD: standard deviation, ES: effect size. *X*^2^ for categorical variables, *t* for continuous variables. ^a^ Cohen’s d, ^b^ Phi, * *p* < 0.05.

**Table 3 nutrients-14-04481-t003:** Differences between USS and NUSS in sports motivation.

Variables	USS (*n* = 249)	NUSS (*n* = 305)	*t* (*p*-Value)	Cohen’s *d*
EM for external regulation, M ± SD	13.15 ± 6.67	12.53 ± 6.85	1.075 (0.283)	0.137
EM for introjected regulation, M ± SD	20.43 ± 5.15	18.76 ± 5.69	3.582 (<0.001) *	0.306
EM for identified regulation, M ± SD	17.93 ± 5.75	17.60 ± 5.75	0.682 (0.496)	0.057
IM to know, M ± SD	22.78 ± 4.72	21.61 ± 5.76	2.566 (0.011) *	0.220
IM to experiment, M ± SD	23.36 ± 4.65	22.17 ± 5.46	2.726 (0.007) *	0.233
IM to achieve results, M ± SD	23.02 ± 4.92	21.83 ± 5.62	2.624 (0.009) *	0.260
Amotivation, M ± SD	11.37 ± 6.62	11.89 ± 6.53	−0.921 (0.357)	−0.079

M: mean, SD: standard deviation; * *p* < 0.05.

**Table 4 nutrients-14-04481-t004:** Differences between USS and NUSS in exercise abuse.

Variables	USS (*n* = 249)	NUSS (*n* = 305)	*t* (*p*-Value)	Cohen’s *d*
Dependance, M ± SD	1.51 ± 1.07	1.19 ± 1.12	3.456 (<0.001) *	0.292
Lack of control, M ± SD	0.83 ± 0.76	0.84 ± 0.78	−0.061 (0.951)	−0.013
Loss of interest, M ± SD	1.83 ± 1.70	1.56 ± 1.50	1.959 (0.051)	0.169
Continuity, M ± SD	1.02 ± 0.85	0.94 ± 0.86	1.036 (0.300)	0.094
Concern, M ± SD	0.77 ± 1.04	0.52 ± 0.89	3.099 (0.002) *	0.259

M: mean, SD: standard deviation, * *p* < 0.05.

**Table 5 nutrients-14-04481-t005:** Differences between USS and NUSS in Adonis Complex.

Variables	USS (*n* = 249)	NUSS (*n* = 305)	*t* (*p*-Value)	Cohen’s *d*
Psychosocial effects of physical appearance, M ± SD	2.73 ± 2.55	3.01 ± 3.26	−1.11 (0.265)	−0.096
Control of physical appearance, M ± SD	2.48 ± 1.95	1.67 ± 1.67	5.287 (<0.001) *	0.448
Concern of physical appearance, M ± SD	2.17 ± 1.79	2.06 ± 1.87	0.746 (0.456)	0.060

M: mean, SD: standard deviation, * *p* < 0.05.

**Table 6 nutrients-14-04481-t006:** Logistic Regression Analysis to classify the set of variables that profile USS.

Variables	B	SE	df	*p*-Value	OR	95% CI for OR	
						Lower	Upper
Sex (men)	0.587	0.189	1	0.002 *	1.799	1.242	2.604
Days per week	0.423	0.078	1	<0.001 *	1.526	1.310	1.778
EM introjected regulation	0.023	0.020	1	0.254	1.023	0.984	1.065
IM to know	0.000	0.029	1	0.989	1.000	0.944	1.059
IM to achieve results	−0.015	0.033	1	0.635	0.985	0.924	1.050
IM to experiment	0.031	0.032	1	0.337	1.031	0.969	1.098
SAS-15 “dependence”	−0.004	0.101	1	0.971	0.996	0.817	1.216
SAS-15 “concern”	0.100	0.112	1	0.370	1.105	0.888	1.376
ACQ Physical appearance	0.182	0.056	1	0.001 *	1.200	1.076	1.338
Constant	−3.360	0.555	1	<0.001 *	0.035		

SE: standard error, OR: odds ratio, CI: confidence interval, significant at * *p* ≤ 0.05.

## Data Availability

The data presented in this study are available upon request from the corresponding author.

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
