# Peer review of "Sports Supplements User Profile Based on Demographic, Sports, and Psychological Variables: A Cross-Sectional Study"

_nutrients, 2022, doi:10.3390/nu14214481_

Round 1

Reviewer 1 Report

The main purpose of the work was to describe the demographic and sports-related characteristics of sports supplement(SS) users but also to incorporate sports motivation (SM), exercise abuse, and MD analyses, to integrate a psychological perspective in the evaluation of the profile of SS users. Therefore, this study aimed to identify the sociodemographic characteristics and anthropometric measures of users of SS (USS) vs. non-users of SS (NUSS). Second, aimed to establish the differences between USS vs. NUSS in terms of sports-related variables, such as status (being an elite athlete vs. a recreational exerciser), days of training per week, hours of exercise per day, the preferred time of the day for training and the psychological variables (SM, exercise abuse and MD). Finally, to be able to profile the USS within demographic, sports and psychological variables, a classification model was developed.

First of all, thank you for the effort you put into your research.

Your research is valuable in terms of its subject and scope.

The topic is relevant, and the study can contribute to the extant literature by providing new theoretical insights and will interest people in the “discipline”.

Introduction: Please, insert new references (2022, ít only one) relevant to the research. 

Discussion: The discussion will always connect to the introduction  you posed and the literature you reviewed.

Given the above:  More relevant references are needed. I suggest you get support from some up-to-date sources.

In addition, I think that the discussion will contribute more to the field with the findings. So, please insert new references (2022) relevant to the research (a few).

Insert a clear limitation, future work and practical limitations of the study.

Conclusion: The conclusion of a research paper is where you wrap up your ideas and leave the reader with a strong final impression.

Author Response

Dear reviewer,

Coauthors and I appreciate your valuable and constructive comments on this manuscript. These comments have been very useful to be able to improve the manuscript. According to the suggestions and comments, we have reviewed the manuscript and incorporated their suggestions in the manuscript. We have also answered to the comments, and our responses are as follows. 

Reviewer #1:

  1. Introduction: Please, insert new references (2022, its only one) relevant to the research. 

Response: We very much appreciate reviewer’s suggestion. We agree with you, even though regarding psychological variables literature is very scarce, we have found important information regarding general profile such as demographic, which is similar of what was found before.  We have updated literature with new references in the introduction (lines: 40-46; 47-48; 52-53; 68-74).

  1. Discussion: The discussion will always connect to the introduction you posed and the literature you reviewed.

Response: Thank you so much for your observation, we agree with you. Therefore, we have also revised the discussion section and made changes in the lines: 276-281; 282-291; 292-299, 341-354). In the case of lines 341 – 354. After a clear revision of different Nutrients journal articles, we have moved the paragraph of limitations to be the last of the discussion sections.

  1. Given the above:  More relevant references are needed. I suggest you get support from some up-to-date sources.

Response: Yes,  as mentioned before we have added up-to-date references that are significant for this study. In the lines: 283-284;292-299; 345-358 changes have been made regarding updated references.

  1. Discussion: Insert a clear limitation, future work, and practical limitations of the study.

Response:  Thank you for pointing this out, we have reviewed changed the limitations from the conclusions to discussion. We have also made some changes, especially in the line 350.

  1. Conclusion: The conclusion of a research paper is where you wrap up your ideas and leave the reader with a strong final impression.

Response:

We fully agree. In the conclusions section now, we have a paragraph with the result of the binary logistic regression model that was done. In the closure we inform that being men, training more days a week and having more concerns regarding physical appearance classifies users of sports supplements.

Reviewer 2 Report

Mera-Zouian L. et al. submitted the article titled: "Sports Supplements user profile based on demographic, sports, and psychological variables. A cross-sectional study."

- First of all, your manuscript is full of grammatical errors and typos. This issue needs to be thoroughly revised.

Abstract:

- „There were significant differences between men and women (50.6% vs. 49.4%, p = 0.004). No significant differences were found between recreational exercisers and elite athletes.” Differences regarding what? Please be precise, especially in the Abstract which should be a clear and a self-sufficient part of your manuscript.

- “Participants that trained more days per week tended to use more SS”. This could mean that there was a positive correlation between the training days/per week and the SS usage or that you had certain groups (example 1-3 days, 3-5 days, 5-7 days training per week). Please be more precise.

- Your whole abstract is focusing on the importance of the psychological factors regarding SS usage and there are zero results regarding that in the Abstract. Revise that.  

Introduction: 

- There is no flow in the text and at some parts you are even losing the main point of your study. Moreover, it is rather too long and it should be more concise regarding the issue you are researching.

-  I would like to see more specifically the psychological aspect of this issue, how it is connected to exercise. You should clarify and elaborate that in a more clear way.

- What does "MD" stand for? What is Adonis Complex? What is bigorexia? This terminology should be clearly explained and you should always name something on the place of the first mention before using an abbreviation.

- “Having reviewed the literature, we did not find sufficient evidence regarding the demographic profile of SS users”. On the other hand, you showed none of the demographic results besides gender and elite/recreational athletes. Moreover, quoting your Abstract: “Most studies have focused on the demographic and sports variables”. So what is it, too much or not enough research regarding demographics. Please elaborate and revise this issue.

Methods:

- The study was conducted during the time period between 2020 and 2022 in which the COVID-19 pandemic has hit the whole world and a lot of countries were during several periods in lockdown. Since part of your questionnaire are instruments which questioned the training sessions (days per week, hour per week), sport motivations, exercise abuse…etc… Is it possible that your results are majorly impacted and biased due to the lockdowns and sport event cancellations? Or did you maybe eliminate this possibility somehow?

Results:

- Please use the symbol ± between mean and SD for presenting continuous variables. On the other hand, leave the whole number (percentage) for the qualitative data.

- Rather use the term “categorical” variable instead of “non-continuous” variable for gender, period of time… and other qualitative variables.

- Are really of your quantitative data normally distributed so you can present them as mean ± SD? Did you use any method to normalize the non-normally distributed variables?

- Since the independent variables in the logistic model are predictors, please use the right terminology regarding them in the textual part (negative or positive). P-value is not the only important part of the log. reg. models.

Discussion:

- The discussion is a bit too long, and similar to the Introduction section, it loses the flow when you read it. Please revise it and put more emphasize on commenting your own results.

Author Response

Dear reviewer,

Coauthors and I appreciate your valuable and constructive comments on this manuscript. These comments have been very useful to be able to improve the manuscript. According to the suggestions and comments, we have reviewed the manuscript and incorporated their suggestions in the manuscript. We have also answered to the comments, and our responses are as follows. 

  1. Abstract:
    1. There were significant differences between men and women (50.6% vs. 49.4%, p = 0.004). No significant differences were found between recreational exercisers and elite athletes.” Differences regarding what? Please be precise, especially in the Abstract which should be a clear and a self-sufficient part of your manuscript.

Response: Thank you very much for your valuable comment, we have reviewed the abstract and have made changes you will be able to find in lines 18-19 and 21-26. Changes made are focused in being more specific regarding what we are informing regarding the psychological variables.

  1. “Participants that trained more days per week tended to use more SS”. This could mean that there was a positive correlation between the training days/per week and the SS usage or that you had certain groups (example 1-3 days, 3-5 days, 5-7 days training per week). Please be more precise.

Response: Thank you for your precise comment, in this sense, this variable es numeric, so what we asked was how many days a week they trained. We have specified in line 21. 

  1. Your whole abstract is focusing on the importance of the psychological factors regarding SS usage and there are zero results regarding that in the Abstract. Revise that.  

Response: Thank you very much, yes absolutely the importance on analyzing the psychological factor is the main purpose of the manuscript, we have added in line 22-23 the results from the binary logistic regression model regarding the psychological aspects.  

  1. Introduction: 

- There is no flow in the text and at some parts you are even losing the main point of your study. Moreover, it is rather too long and it should be more concise regarding the issue you are researching.

Response: Yes, we agree with you, it’s a bit too long. The main idea of the introduction has always been to report what was found regarding demographic, sports, and psychological variables. From one paragraph to another, it was confusing and lacking a little sense due to the amount of information. Therefore, the introduction has been revised. Parragraphs 2,3,4 and 5 are revised and written with some new references too.

  1. -  I would like to see more specifically the psychological aspect of this issue, how it is connected to exercise. You should clarify and elaborate that in a clearer way.

Response: Thank you so much for your precious suggestion. In this sense, there is a lack of evidence regarding the psychological factors and the use of sports supplements. This matter has not been studied in full previously. Regarding exercise and the use of SS, we have considered the variables of type of athletes and time of training.

  1. - What does "MD" stand for? What is Adonis Complex? What is bigorexia? This terminology should be clearly explained, and you should always name something on the place of the first mention before using an abbreviation.

Response: You are right, first there was an escape from the MD not being named, sorry for that. Regarding the definitions, we have added just some words to explain this. It is important for the readers to have definitions clear (lines 61 and 63).

  1. - “Having reviewed the literature, we did not find sufficient evidence regarding the demographic profile of SS users”. On the other hand, you showed none of the demographic results besides gender and elite/recreational athletes. Moreover, quoting your Abstract: “Most studies have focused on the demographic and sports variables”. So what is it, too much or not enough research regarding demographics. Please elaborate and revise this issue.

Response: Thank you very much for your concern. Regarding psychological variables not enough scientific evidence was found, and reports from research of the profile of sports supplements have been very clear regarding the inconclusive that previous investigations have been. One of the main reasons are the lack if defining supplements.

  1. Methods:

-he study was conducted during the time period between 2020 and 2022 in which the COVID-19 pandemic has hit the whole world and a lot of countries were during several periods in lockdown. Since part of your questionnaire are instruments which questioned the training sessions (days per week, hour per week), sport motivations, exercise abuse…etc… Is it possible that your results are majorly impacted and biased due to the lockdowns and sport event cancellations? Or did you maybe eliminate this possibility somehow?

Response:  Thank you very much for this precise suggestion. We strongly agree that the moment where this study took place, should be considered. We have added this matter in the limitations our can present.

  1. Results:

- Please use the symbol ± between mean and SD for presenting continuous variables. On the other hand, leave the whole number (percentage) for the qualitative data.

Response:  Thank you very much for your suggestion,  we agree and this has been changed in the manuscript from M (SD) to be M±SD.

  1. - Rather use the term “categorical” variable instead of “non-continuous” variable for gender, period of time… and other qualitative variables.

Response:  Thank you very much for your valuable comment. Important to say again that the days of training is a numeric variable so we have used M (SD). For the categorical we have revised the text and corrected following your suggestions.  

  1. - Are really of your quantitative data normally distributed so you can present them as mean ± SD? Did you use any method to normalize the non-normally distributed variables?

Response: Yes, we have. This has been explained in the statistical analysis segment (Data normality was tested by using the Shapiro–Wilk and Kolmogorov–Smirnov tests) (line 191).

  1. - Since the independent variables in the logistic model are predictors, please use the right terminology regarding them in the textual part (negative or positive). P-value is not the only important part of the log. reg. models.

Response: Thank you very much, the independent variable is the predictor, in our case the use of Sports Supplements (SS). In lines 246-247 you may see we have “This model correctly classified 66.8% of the USS with a significant Chi-square result (X2 = 78.595, p < 0.01). The r2 of Nagelkerke was low (r2 = 0.177)”. Also, when we have an OR over 1 we have a positive power or classification regarding the variable we a analyzing. In this case we have also added the correct terminology/ Line 256.

  1. Discussion:

- The discussion is a bit too long, and like the Introduction section, it loses the flow when you read it. Please revise it and put more emphasize on commenting your own results.

Response: Thank you very much for your valuable comment, this has been the hard part. But after changing the introduction in the matter. Of. Flow and not so long we have done the same with the conclusions. Changes have been made in: 272-282, 284-291, 289-292, 292-300, 346-359.

***Regarding the comment of the English, this manuscript was submitted to the editing services of MDPI before it was sent to the journal. Afterwards we did some changes, and this may have affected.   We have spoken to the editing services, and they have suggested to in revise and incorporate as before regarding the changes.  We have revised the manuscript English again. In case this is still an issue, we kindly suggest a chance to be able to send the manuscript to English editing services for a response. 

PLEASE SEE ATTACHMENT REGARDING THE CHANGES ON THE MANUSCRIPT. We have downloaded again the template for this matter, to be able to have the lines numbered. 

Round 2

Reviewer 2 Report

Dear Authors,

You have elaborated an answer to my every comment and you have significantly improved the quality of your manuscript.

Author Response

Thank you very much. They were all very welcomed.